# Yohimbine, an α2-Adrenoceptor Antagonist, Suppresses PDGF-BB-Stimulated Vascular Smooth Muscle Cell Proliferation by Downregulating the PLCγ1 Signaling Pathway

**DOI:** 10.3390/ijms23148049

**Published:** 2022-07-21

**Authors:** Chih-Wei Chiu, Cheng-Ying Hsieh, Chih-Hao Yang, Jie-Heng Tsai, Shih-Yi Huang, Joen-Rong Sheu

**Affiliations:** 1Graduate Institute of Medical Sciences, College of Medicine, Taipei Medical University, Taipei 110, Taiwan; sky52250@hotmail.com; 2Department of Pharmacology, School of Medicine, College of Medicine, Taipei Medical University, Taipei 110, Taiwan; hsiehcy@tmu.edu.tw (C.-Y.H.); chyang@tmu.edu.tw (C.-H.Y.); a19851102@tmu.edu.tw (J.-H.T.); 3School of Nutrition and Health Sciences, Taipei Medical University, Taipei 110, Taiwan; 4Graduate Institute of Metabolism and Obesity Sciences, Taipei Medical University, Taipei 110, Taiwan; 5Center for Reproductive Medicine & Sciences, Taipei Medical University Hospital, Taipei 110, Taiwan

**Keywords:** neointimal formation, yohimbine, vascular smooth muscle cells, PLCγ1

## Abstract

Yohimbine (YOH) has antiproliferative effects against breast cancer and pancreatic cancer; however, its effects on vascular proliferative diseases such as atherosclerosis remain unknown. Accordingly, we investigated the inhibitory mechanisms of YOH in vascular smooth muscle cells (VSMCs) stimulated by platelet-derived growth factor (PDGF)-BB, a major mitogenic factor in vascular diseases. YOH (5–20 μM) suppressed PDGF-BB-stimulated a mouse VSMC line (MOVAS-1 cell) proliferation without inducing cytotoxicity. YOH also exhibited antimigratory effects and downregulated matrix metalloproteinase-2 and -9 expression in PDGF-BB-stimulated MOVAS-1 cells. It also promoted cell cycle arrest in the initial gap/first gap phase by upregulating p27Kip1 and p53 expression and reducing cyclin-dependent kinase 2 and proliferating cell nuclear antigen expression. We noted phospholipase C-γ1 (PLCγ1) but not ERK1/2, AKT, or p38 kinase phosphorylation attenuation in YOH-modulated PDGF-BB-propagated signaling pathways in the MOVAS-1 cells. Furthermore, YOH still inhibited PDGF-BB-induced cell proliferation and PLCγ1 phosphorylation in MOVAS-1 cells with α2B-adrenergic receptor knockdown. YOH (5 and 10 mg/kg) substantially suppressed neointimal hyperplasia in mice subjected to CCA ligation for 21 days. Overall, our results reveal that YOH attenuates PDGF-BB-stimulated VSMC proliferation and migration by downregulating a α2B-adrenergic receptor–independent PLCγ1 pathway and reduces neointimal formation in vivo. Therefore, YOH has potential for repurposing for treating atherosclerosis and other vascular proliferative diseases.

## 1. Introduction

Abnormal proliferation of vascular smooth muscle cells (VSMCs) is among the critical factors associated with vascular diseases such as atherosclerosis and restenosis [1,2]. In response to arterial wall injury, VSMCs proliferate and migrate from the medial layer to the intimal layer, which is a hallmark of neointimal hyperplasia [3]. Numerous proinflammatory cytokines and growth factors participate in the progression of neointimal hyperplasia, and platelet-derived growth factor (PDGF)-BB plays a major role in the proliferation and migration of VSMCs [4]. Therefore, therapeutic strategies against PDGF-BB-mediated VSMC proliferation and migration are urgently required for treating and preventing vascular proliferative diseases [5].

VSMCs respond to PDGF-BB stimulation, which activates downstream signaling pathways and then drives cell cycle progression, cell proliferation, and cell migration [6]. These biological processes are triggered by the binding of PDGF-BB to its receptor, platelet-derived growth factor receptor (PDGFR)-β, which activates mitogenic signals through the phosphorylation of tyrosine residues. The phosphorylated PDGFR-β further interacts with multiple proteins that contain Src homology 2 (SH2) domains, including phosphatidylinositol 3-kinase (PI3K), tyrosine phosphatase SHP-2, and phospholipase Cγ1 (PLCγ1) [1]. In addition to the primary messengers, PDGF-BB propagates multiple signaling cascades such as mitogen-activated protein kinase (MAPK) and AKT phosphorylation [6,7,8]; MAPKs and AKT play a key role in the regulation of cell proliferation and migration. Studies have shown that PLCγ1 knockdown significantly reduced PDGF-BB-induced VSMC proliferation and migration. In vivo data also demonstrated that continuous administration of PLCγ1 siRNA to mice markedly reduced neointimal formation induced by carotid artery ligation [9]. In addition, a study reported that targeting the activated PI3K and PLCγ1 signaling in vivo could abolish PDGF-BB-induced neointimal formation after vascular injury [10].

Yohimbine is an indole alkaloid derived from the bark of the African tree *Pausinystalia yohimbe* and is a selective antagonist of α2-adrenergic receptors. It has been used in the treatment of erectile dysfunction for years. Yohimbine (YOH) can induce penile vascular smooth muscle contraction by selectively blocking presynaptic α2-adrenergic receptors [11,12]. Moreover, YOH has various pharmacological properties, including antioxidant [13,14], anti-inflammatory, and antianxiety properties [15]. In addition to these pharmacological features, YOH can regulate endocrine functions [16] and can serve as an analgesic [17]. Studies have reported that YOH possesses antiproliferative effects against breast cancer [18] and pancreatic cancer [19]. Moreover, α2-adrenergic receptors can modulate vasoconstriction in vascular smooth muscle and have been reported to play a potential role in macrophage function and atherosclerosis promotion [20,21]. Three subtypes of α2-adrenergic receptors exist, namely α2A, α2B, and α2C, and each of them plays a different role in cardiovascular regulation [22,23,24]. Among these three subtypes, the α2B-adrenergic receptor has been observed to play a potential role in the regulation of VSMC proliferation [25]. Therefore, as a selective α2-adrenergic receptor antagonist, YOH may have therapeutic efficacy against neointimal hyperplasia through its anti-proliferative property in VSMCs. However, no study has determined the effects of YOH on PDGF-BB-stimulated proliferation in VSMCs. Accordingly, the present study conducted in vivo and in vitro experiments to investigate the inhibitory effects of YOH on PDGF-stimulated VSMC proliferation and determined the mechanisms underlying these effects. The anti-proliferative ability of YOH in VSMCs may be advantageous to treating vascular proliferative diseases.

## 2. Results

### 2.1. Effects of YOH on PDGF-BB-Induced Proliferation and Cell Cycle Progression in MOVAS-1 Cells

We first determined the effects of YOH on PDGF-BB-induced VSMC proliferation by using the MTT assay, a typical assay to measure cellular metabolic activity as an indicator of cell viability [26]. As illustrated in Figure 1B(a), PDGF-BB (10 ng/mL) stimulated the MOVAS-1 cells to proliferate by more than 50%, and YOH pretreatment at 5, 10, and 20 μM substantially inhibited PDGF-BB-stimulated MOVAS-1 cell proliferation, with the corresponding proliferation rates being approximately 22.0%, 17.0%, and 34.0%, respectively. Furthermore, YOH administered at the indicated concentrations (5–20 μM) exhibited no cellular toxicity (Figure 1B(b)). In addition to the MTT assay results, microscopic observation revealed that YOH treatment markedly reduced the proliferation of the PDGF-BB-stimulated MOVAS-1 cells (Figure 1C).

We next applied flow cytometry to examine the influence of YOH on cell cycle progression in the PDGF-BB-stimulated MOVAS-1 cells. The results revealed significant increases in the percentage of PDGF-BB-treated (10 ng/mL) VSMCs in the synthetic (S) phase (from 9.1 ± 0.2% to 17.4 ± 0.9%; *p* < 0.001) and second gap (G2)/mitosis (M) phase (from 22.8 ± 0.9% to 30.6 ± 1.4%; *p* < 0.001); however, the percentage of PDGF-BB-treated (10 ng/mL) MOVAS-1 cells in the initial gap (G0)/first gap (G1) phase decreased from 68.1 ± 0.8% to 52 ± 1.8% (Figure 1D(a–e)). YOH administration engendered a notable accumulation of cells in the G0/G1 phase but induced a reduction of those in the G2/M phase compared with the control cells, which received solvent treatment (G0/G1 phase, 52 ± 1.8% vs. 55.4 ± 0.8%; *p* < 0.01; G2/M phase, 30.6 ± 1.4% vs. 27.9 ± 0.9%; *p* < 0.01; Figure 1D). These results suggest that YOH can arrest cell cycle progression in the G0/G1 phase and then affect cell proliferation in PDGF-BB-stimulated MOVAS-1 cells.

### 2.2. YOH Inhibits PDGF-BB-Stimulated Migration as Well as MMP-2 and MMP-9 Expression in MOVAS-1 Cells

To investigate whether YOH influences cell migration in response to PDGF-BB in VSMCs, we conducted a wound healing assay to evaluate the migratory capacity of the cells [27]. We also examined the expression levels of MMP-2 and MMP-9—two critical metalloproteinases in vascular remodeling and VSMC migration [28,29]—in PDGF-BB-stimulated MOVAS-1 cells treated with or without YOH. As displayed in Figure 2A(a–e), YOH pretreatment (20 μM) significantly inhibited PDGF-BB-induced MOVAS-1 cell migration. Moreover, the expression levels of MMP-2 and MMP-9 were substantially increased in the PDGF-BB-stimulated (10 ng/mL) MOVAS-1 cells, and the preincubation of the cells with YOH (10 and 20 μM) suppressed the expression of MMP-2 and MMP-9 in a concentration-dependent manner (Figure 2B(a,b)). YOH pretreatment also exhibited strong migration inhibition activity in the PDGF-BB-stimulated MOVAS-1 cells.

### 2.3. YOH Modulates the Expression of Cell-Cycle-Associated Proteins in VSMCs Treated with PDGF-BB

As presented in Figure 3A,B, PDGF-BB administration (10 ng/mL) significantly upregulated CDK2 and PCNA expression in the MOVAS-1 cells, and YOH pretreatment (20 μM) substantially attenuated PDGF-BB-induced CDK2 and PCNA expression. We further investigated the protein expression of negative cell cycle modulators such as p27Kip1 and the tumor suppressor p53 in YOH-treated MOVAS-1 cells. The protein expression levels of p27Kip1 and p53 were significantly induced after YOH administration (20 μM; Figure 3C,D). These results suggest that YOH caused G0/G1 cell cycle arrest in the PDGF-BB-stimulated MOVAS-1 cells by modulating these cell-cycle-associated proteins.

### 2.4. Mechanisms Underlying the Inhibitory Effects of YOH on PLCγ1 Phosphorylation in PDGF-BB-Treated MOVAS-1 Cells

To clarify the mechanisms underlying the effects of YOH against PDGF-BB-stimulated VSMC activation, we examined the expression of PDGF-mediated signaling pathways such as the MAPK, AKT, and PLCγ1 pathways [7,8]. We observed that PDGF-BB treatment (10 ng/mL) significantly upregulated p38MAPK, ERK1/2, AKT, and PLCγ1 phosphorylation in the MOVAS-1 cells (Figure 4A–C). Nevertheless, YOH administration at 20 μM substantially attenuated only PLCγ1 phosphorylation in the PDGF-BB-stimulated MOVAS-1 cells (Figure 4D). PDGF-BB-induced ERK1/2, AKT, and p38MAPK phosphorylation in the MOVAS-1 cells was not affected by YOH. Moreover, as illustrated in Figure 4E, YOH treatment (20 μM) also inhibited the expression of PLCγ1-derived phosphorylation of Ser residues [30] in PDGF-BB-stimulated MOVAS-1 cells.

### 2.5. Downregulation of Cell Proliferation and PLCγ1 Phosphorylation by YOH Is Independent of Its Known α2-Antagonism in MOVAS-1 Cells Stimulated by PDGF-BB

YOH served as a selective α2-adrenergic receptor antagonist. The α2-adrenergic receptor belongs to a family of G protein-coupled receptors and consists of three highly homologous subtypes: α2A-, α2B-, and α2C [31]. Among them, the α2B-adrenergic receptor subtype has been reported to be involved in VSMC proliferation [20,25]. Accordingly, we performed α2B-adrenergic receptor knockdown in the MOVAS-1 cells (Figure 5A) to determine whether the inhibitory effects of YOH on cell proliferation and PLCγ1 proliferation in the PDGF-BB-stimulated MOVAS-1 cells were associated with its α2-receptor antagonism. As displayed in Figure 5B, YOH treatment still substantially inhibited PDGF-BB-induced cell proliferation in MOVAS-1 cells with an α2B-adrenergic receptor knockdown. In addition, YOH attenuated both PLCγ1 phosphorylation and downstream Ser residue phosphorylation in PDGF-BB-stimulated MOVAS-1 cells with an α2B-adrenergic receptor knockdown (Figure 5C,D).

### 2.6. YOH Suppresses Neointimal Hyperplasia in a Mouse Model of CCA Ligation

We adapted a mouse model of CCA ligation [32] to conduct in vivo experiments to assess the possible inhibitory effects of YOH on neointimal hyperplasia. The mice were cotreated with or without YOH (5 and 10 mg/kg/d, injected intraperitoneally) for 21 days, and CCA cross sections were then subjected to H&E staining. As presented in Figure 6A, YOH treatment notably alleviated neointimal hyperplasia in the model in a concentration-dependent manner. Statistics regarding the intima–media ratio (I/M ratio) also revealed a significant reduction in neointimal hyperplasia in the YOH-treated mice compared with the solvent control group (from 3.7 ± 0.6 to 1.5 ± 0.3; *p* < 0.001; Figure 6B).

## 3. Discussion

The present study demonstrates that YOH, a conventional drug that is commonly used for erectile dysfunction, can be used to treat neointimal hyperplasia. In fact, YOH was also reported to decrease body fat by upregulating lipolysis and leptin content, and shows anorectic activity to reduce body weight [33,34]. Drug reproposing of YOH seems to be attractive in clinical application. VSMC proliferation and migration are responsible for the pathogenesis of neointimal formation in atherosclerosis or restenosis after balloon angioplasty. Our data indicate that YOH treatment significantly suppressed PDGF-BB-stimulated VSMC proliferation and migration by inhibiting PLCγ1 phosphorylation; YOH treatment also caused cell cycle arrest in the G0/G1 phase. In addition, in vivo experiments revealed that YOH substantially attenuated neointimal hyperplasia in a mouse model of CCA ligation (Figure 6C).

VSMC proliferation is critical in the pathogenesis of atherosclerosis and associated vascular diseases [35,36]. Abnormal VSMC proliferation can be induced by a multitude of mitogenic factors, among which PDGF-BB is the most potent cell proliferation stimulator that acts through the activation of PDGFR signaling and further signal propagation. Consequently, mitigating PDGF-BB-induced VSMC proliferation is advantageous for treating cardiovascular diseases. In this study, we found that YOH administration (5–20 μM) significantly suppressed PDGF-BB-induced cell proliferation without inducing cellular toxicity in MOVAS-1 cells. In addition to VSMC proliferation, VSMC migration is a crucial characteristic in the pathogenesis of atherosclerosis [37]. To move through the matrix, VSMCs produce a family of zinc-dependent enzymes called MMPs to degrade various extracellular proteins. VSMCs produce MMPs including MMP-2 and MMP-9 in response to various stimuli, including cytokines, growth factors, and mechanical stress, and MMP-2 and MMP-9 are highly expressed in atherosclerotic lesions and intimal hyperplasia [27,38]. MMP downregulation efficiently alleviates VSMC migration and subsequent vascular remodeling in animal models of vascular injury [28,29]. Our data demonstrate that YOH treatment significantly reduced the expression of both MMP-2 and MMP-9 in, and the migratory capacity of, PDGF-BB-stimulated MOVAS-1 cells. Similarly, a previous study reported that YOH inhibited breast cancer cell proliferation, migration, and invasion [18].

In a normal medial layer, VSMCs exhibit a quiescent and contractile phenotype, which remains in the G0 phase, a non-proliferative cell cycle phase [10]. In response to vascular injury, VSMCs undergo cell cycle transition from a contractile phenotype to an active or synthetic phenotype by entering the G1 phase [39]. Subsequently, VSMCs in the S phase produce substances for DNA replication, and they finally transition from the G2 phase to the M phase after DNA replication. In this study, we found that YOH-treated MOVAS-1 cells lost their proliferative capability and remained the G0/G1 phase. In addition to conducting flow cytometry, we evaluated the protein expression levels of CDK2, PCNA, p27Kip1, and p53 to clarify the possible mechanism through which YOH regulates cell cycle progression. We noted that YOH administration significantly attenuated PDGF-BB-induced CDK2 and PCNA upregulation in the VSMCs. A previous study revealed that CRISPR/Cas9-mediated knockout of CDK2 induced G0/G1 arrest and apoptosis in A375 melanocytes [40], and several other studies have shown a substantial downregulation of CDK2 in G0/G1-arrested VSMCs [39,41,42]. This, thus, explains why the expression of PCNA, a classic biomarker of proliferative cells, was suppressed in the YOH-treated MOVAS-1 cells. By contrast, p27Kip1 and p53, which are negative regulators of cell cycle progression, were both upregulated in the YOH-treated MOVAS-1 cells. Previous studies have indicated that overexpression of p27Kip1, a member of the KIP/CIP family, inhibits cellular proliferation and leads to cell cycle arrest in the G0/G1 phase [43,44]. In addition to p27Kip1, p53 was also reported to regulate VSMC survival through the modulation of downstream target genes. Moreover, p53 pathway activation has been reported to promote cell cycle arrest in the G0/G1 phase and attenuate consequent neointimal hyperplasia [45,46]. The inhibitory effects of YOH on PDGF-BB-induced VSMC proliferation may be associated with the upregulation of p27Kip1 and p53-mediated G0/G1 phase arrest and the subsequent reduction of CDK2 and PCNA expression levels.

PDGFR signaling is activated by PDGF-BB binding, which leads to the phosphorylation of PDGFR-β as well as the phosphorylation and activation of several downstream molecules such as ERK1/2, p38, AKT, and PLCγ1 [47,48]. In this study, only PLCγ1 phosphorylation was downregulated by YOH in the PDGF-BB-stimulated VSMCs. PDGF-BB-phosphorylated ERK1/2, p38, and AKT were unaffected by the administration of YOH. Previous research indicated that PDGF-modulated cell cycle progression is notably mediated by PLCγ1 and PI3K, which differentially regulate p27Kip1 and cyclin D1 [10]. The inhibition of PDGF-induced PLCγ1 signaling activation in mice results in a substantial attenuation of neointima hyperplasia [9]. However, a recent study indicated that the α2B-adrenergic receptor may exhibit proliferative effects by promoting the activity of growth factors and their receptors in VSMCs [25]. Accordingly, we performed α2B-adrenergic receptor knockdown in the VSMCs to clarify the association of PLCγ1 phosphorylation downregulation and α2B-adrenergic receptor antagonism by YOH in MOVAS-1 cells stimulated by PDGF-BB. YOH treatment still significantly inhibited PDGF-BB-induced cell proliferation and PLCγ1 phosphorylation in the MOVAS-1 cells with α2B-adrenergic receptor knockdown. Accordingly, the inhibitory effects of YOH, a well-known antagonist of α2-adrenergic receptors, on PDGF-BB-stimulated VSMC proliferation might be at least partially accomplished through the downregulation of the PLCγ1-dependent signaling pathway. On the other hand, vasoconstriction is known to be critical in the progression of atherosclerosis and may participate atherosclerotic plaque rupture. Modulation of α2-adrenergic receptor and leptin were both known to inhibit angiotensin-II-stimulated vasoconstriction [49,50]. The therapeutic capacity of YOH on atherosclerotic diseases may be not only through its anti-proliferative property but also by the regulation of vasoconstriction.

In conclusion, this study demonstrates for the first time that YOH, a selective antagonist of α2-adrenergic receptors, suppresses PDGF-BB-induced cell proliferation and migration in VSMCs and the consequent neointimal hyperplasia in a mouse model of CCA ligation. PLCγ1 signaling pathway downregulation may play a crucial role in the inhibitory effects of YOH on the proliferation of PDGF-BB-stimulated VSMCs. Accordingly, YOH may have therapeutic potential for repurposing for treating VSMC-associated cardiovascular diseases.

## 4. Materials and Methods

### 4.1. Materials

Yohimbine hydrochloride was purchased from Cayman Chem (Ann Arbor, MI, USA). Fetal bovine serum (FBS), Dulbecco’s modified Eagle’s medium (DMEM), L-glutamine/penicillin/streptomycin, trypsin (0.25%), and anti-α-tubulin monoclonal antibody (mAb) were purchased from Thermo Fisher Scientific (Waltham, MA, USA). Moreover, recombinant Human PDGF-BB (carrier-free) was purchased from BioLegend (San Diego, CA, USA). RNase A was purchased from New England Biolabs (Ipswich, MA, USA) and 3-(4,5-dimethylthiazol-2-yl)-2,5-diphenyltetrazolium bromide (MTT) was purchased from Merck (Branchburg, NJ, USA). Horseradish-peroxidase-conjugated donkey antirabbit IgG and sheep antimouse IgG were purchased from Amersham (Buckinghamshire, UK). Anti-phospho-ERK1/2 (Thr^202^/Tyr^204^)/(Thr^185^/Tyr^187^) polyclonal antibody (pAb) was purchased from GeneTex (Irvine, CA, USA), and anti-matrix metalloproteinase (MMP)-2 pAb was purchased from BioVision (Milpitas, CA, USA). Anti-MMP-9 pAb was purchased from Abcam (Cambridge, UK). Anti-phospho-p38 MAPK (Thr^180^/tyr^182^) pAb and anti-p27Kip1 pAb were purchased from Affinity Biosciences (Cincinnati, OH, USA). Anti-phospho-AKT mAb, anti-PCNA mAb, anti-CDK2 mAb, anti-phospho-PLCγ1 (Tyr^783^) pAb, and anti-phospho-(Ser) PKC pAb were purchased from Cell Signaling (Beverly, MA, USA). Santa Cruz Biotechnology (Dallas, TX, USA)-produced anti-p53 mAb and anti-α2B-adrenergic receptor mAb were utilized. Enhanced chemiluminescence Western blotting detection reagent and polyvinylidene difluoride (PVDF) membranes were from GE Healthcare Life Sciences (Waukesha, WI, USA). Yohimbine hydrochloride was dissolved in dimethyl sulfoxide (DMSO) and stored at −20 °C until use.

### 4.2. VSMC Cell Culture and Lentivirus Infection

A murine vascular smooth muscle cell line (MOVAS-1) was purchased from ATCC (Manassas, VA, USA; ATCC number: CRL-2797). Cells were cultured in DMEM supplemented with 10% FBS, penicillin G (100 units/mL), streptomycin (100 mg/mL), and glutamine (2 mM) at 37 °C in a humidified atmosphere of 5% CO_2_. Moreover, α2B-adrenergic receptor shRNA (m) lentiviral particles purchased from Santa Cruz Biotechnology were added to the culture. The MOVAS-1 cells were seeded in a six-well plate at a density of 1.5 × 10^5^ cells/well and then transfected with shRNA lentiviral particles mixed with Polybrene in Opti-MEM (Thermo Fisher Scientific, Waltham, MA, USA) for 24 h. To select stable clones, the medium was replaced with fresh puromycin-containing medium (10 μg/mL) every 3 days.

### 4.3. Proliferation Assay (MTT Assay)

MOVAS-1 cells were seeded in 24-well plates (4 × 10^4^ cells/well) and cultured in DMEM with 10% FBS for 24 h. The medium was then replaced with serum-free medium. The cells were pretreated with YOH (5, 10, or 20 μM) for 45 min and then stimulated with or without PDGF-BB (10 ng/mL) for 48 h. Cell viability was measured using a colorimetric assay based on the ability of mitochondria in viable cells to reduce MTT into purple formazan, as described previously [26]. The cell number index was derived as follows: absorbance of treated cells/control cells × 100%.

### 4.4. Wound Healing Assay

MOVAS-1 cells were cultured in six-well plates and grown up to 90% confluence. The cell monolayers were scratched using a 1 mL pipette tip, and the medium was subsequently replaced with serum-free medium after scratching damage. The cells were preincubated with YOH at indicated concentrations (10 or 20 μM) for 45 min and then stimulated with PDGF-BB (10 ng/mL) for 24 h. Images of the migrated area were captured by a microscope (Nikon, Tokyo, Japan) and analyzed using ImageJ.

### 4.5. Cell Cycle Analysis

In the cell cycle analysis, starved MOVAS-1 cells (1 × 10^6^ cells/dish) were pretreated with YOH (10 or 20 μM) for 45 min and then stimulated with or without PDGF-BB (10 ng/mL) for 18 h. After 18 h, the cells were collected using trypsin, washed with cold PBS, and then fixed in ice-cold 70% ethanol overnight. The cell samples were then washed with PBS and resuspended in a PI solution (PI, 80 μg/mL; RNase, 0.2 mg/mL; and Triton X-100, 0.1%) for 30 min and subjected to a flow cytometry test (Attune NxT, Thermo Fisher Scientific Inc., Waltham, MA, USA).

### 4.6. Western Blotting

The cell lysates were extracted using RIPA buffer (containing enzyme inhibitors including aprotinin, PMSF, leupeptin, NaF, sodium orthovanadate, and sodium pyrophosphate) and then subjected to a gel electrophoresis system. The protein samples were loaded and separated on sodium dodecyl sulfate–polyacrylamide gel. After 1.5 h, the separated proteins were transferred to PVDF blotting membranes by semi-dry electrophoretic transfer cell (Bio-Rad; Hercules, CA, USA). The transferred membranes were incubated with BlockPRO blocking buffer (Ruiguang Rd., Neihu Dist., Taipei, Taiwan) for 1 h and then with different primary antibodies for 2 h, after which they were incubated with a secondary antibody for 1 h. An ECL system was used to detect immune-reactive bands in this study. Densitometry was performed to quantify protein bands by using Visionworks for Windows (San Antonio, TX, USA).

### 4.7. Mouse Model of Common Carotid Artery Ligation Model

A mouse model of common carotid artery (CCA)-ligation–induced neointimal hyperplasia was applied as previously described, with some modifications [32]. This protocol was confirmed and approved by IACUC of Taipei Medical University, No. LAC-2018-0063 (1 May 2018 to 30 April 2019). Briefly, male C57BL/6 mice (10-week-old) purchased from BioLASCO (Taipei, Taiwan) were anesthetized using a gas mixture containing 75% air and 3% isoflurane maintained in 25% oxygen. The left CCA was ligated with a 6-0 silk suture near the carotid bifurcation. YOH (5 or 10 mg/kg/day) was injected intraperitoneally into the C57BL/6 mice for 21 days after CCA ligation. After 21 days, the mice were euthanized and the vessels were captured. The vessels were fixed with 4% formaldehyde and embedded in paraffin. The cross sections were stained with hematoxylin and eosin (H&E), and the thicknesses of the intimal and medial layers of the arterial sections were analyzed using ImageJ.

### 4.8. Statistical Analysis

The results are presented as the mean ± standard error (S.E.M.) and the number of independent experiments (*n*). Data were first assessed using one-way analysis of variance (ANOVA). If the one-way ANOVA revealed significant differences among the group means, then the Newman–Keuls method was applied subsequently for comparison. A *p* value of <0.05 was considered to indicate a statistically significant difference.

## Figures and Tables

**Figure 1 ijms-23-08049-f001:**
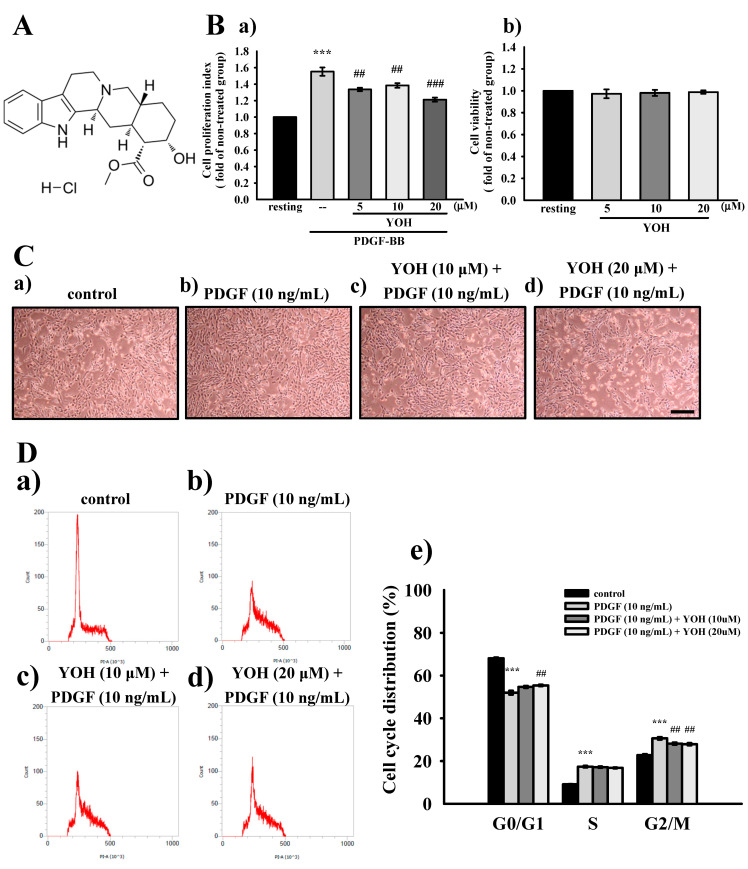
Effects of YOH on PDGF-BB-induced proliferation and cell cycle progression in MOVAS-1 cells. (**A**) Chemical structure of YOH. (**B**) Starved MOVAS-1 cells (4 × 10^4^ cells/well) were pretreated with YOH (5, 10, and 20 μM) for 45 min and then treated with PDGF-BB (10 ng/mL) for 48 h, and the cell cycle progression was monitored for 18 h. The cell proliferation index (**a**) and cell viability (**b**) were measured as described in the Materials and Methods section. (**C**) Images of the morphology of MOVAS-1 cells stimulated without (**a**) or with (**b**) PDGF-BB (10 ng/mL) after pretreatment with YOH at (**c**) 10 or (**d**) 20 μM, the black bar indicates 100 μm. (**D**) Cells were pretreated without (**a**) or with (**b**) solvent control (0.1% DMSO), (**c**) 10 μM YOH, or (**d**) 20 μM YOH and then stimulated with PDGF-BB (10 ng/mL) for 18 h. The samples were collected by trypsinization and then subjected to flow cytometry. The complied statistical data are included (**e**). *** *p* < 0.001, compared with the resting group; ^##^
*p* < 0.01 and ^##^^#^
*p* < 0.001 compared with the solvent control group. Data are presented as the mean ± S.E.M. (*n* = 4).

**Figure 2 ijms-23-08049-f002:**
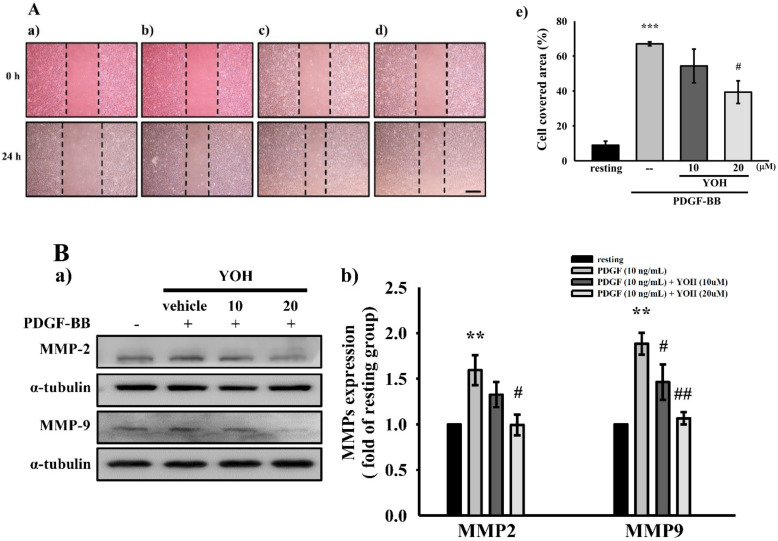
Effects of YOH on cell migration and MMP-2 and MMP-9 expression in PDGF-BB-stimulated MOVAS-1 cells. (**A**) MOVAS-1 cell monolayers were scratched and then preincubated without (**a**) or with 0.1% DMSO (**b**) or YOH ((**c**) 10 or (**d**) 20 μM) for 45 min and then treated with PDGF (10 ng/mL) for 24 h. The quantified wound closure area (%) is shown in (**e**). (**B**) (**a**) Protein expression levels of MMP-2 and MMP-9 were evaluated through Western blotting. (**b**) Complied statistical data are presented on the right. ** *p* < 0.01, *** *p* < 0.001 compared with the resting group; ^#^
*p* < 0.05, ^##^
*p* < 0.01 compared with the positive group. Data are presented as the mean ± S.E.M. (*n* = 4).

**Figure 3 ijms-23-08049-f003:**
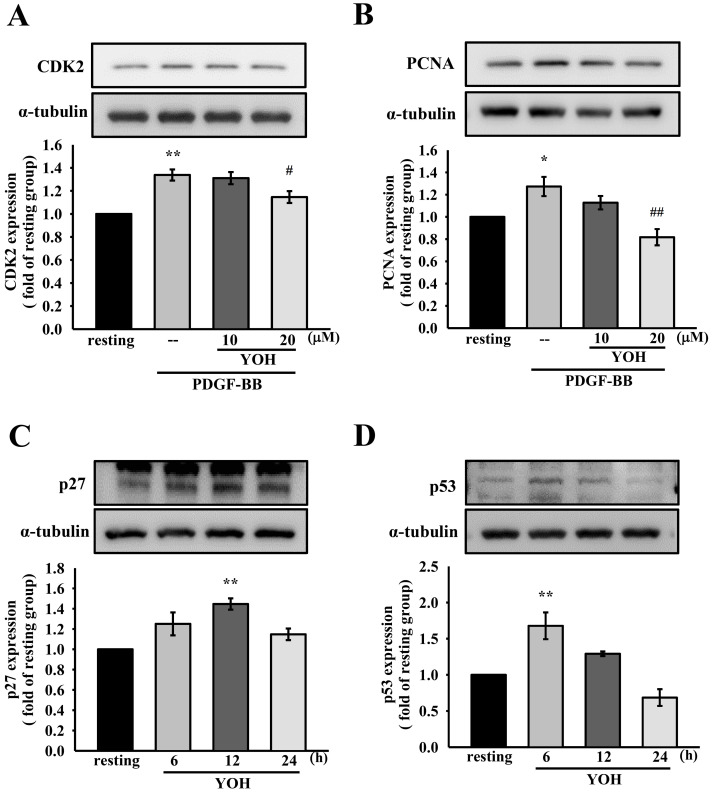
YOH suppresses expression of PDGF-BB-induced CDK2 and PCNA and upregulates expression of p27Kip1 and p53 in MOVAS-1 cells. MOVAS-1 cells were incubated with YOH (10 or 20 μM) for 45 min and treated with (**A**,**B**) or without (**C**,**D**) PDGF-BB stimulation (10 ng/mL). Protein expression levels of (**A**) CDK2, (**B**) PCNA, (**C**) p27Kip1, and (**D**) p53 were determined through Western blotting, as described in the Materials and Methods section. * *p* < 0.05, ** *p* < 0.01 compared with the resting group; ^#^
*p* < 0.05, ^##^
*p* < 0.01 compared with the positive group. Data are presented as the mean ± S.E.M. (*n* = 3).

**Figure 4 ijms-23-08049-f004:**
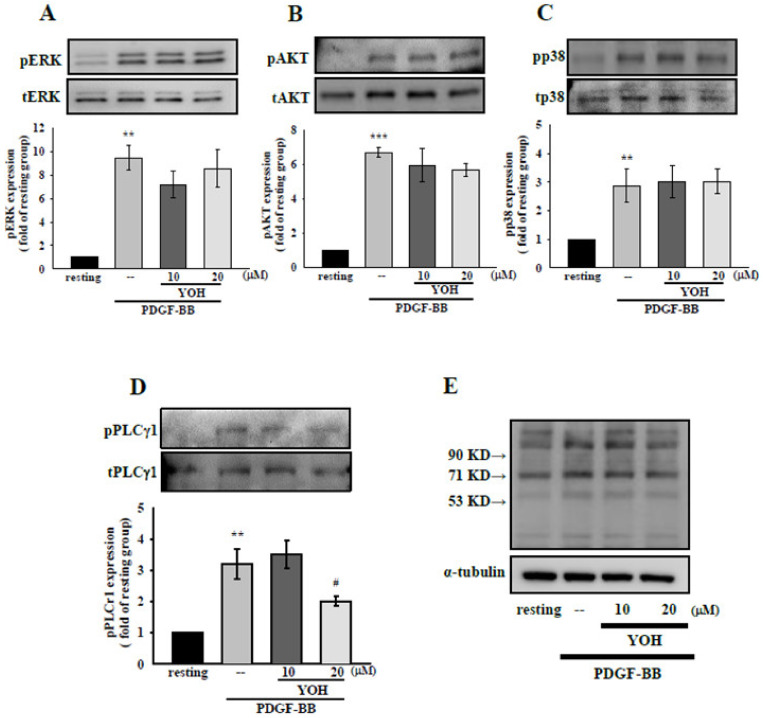
Effects of YOH on ERK1/2, p38, AKT, and PLCγ1 pathway activation in PDGF-BB-stimulated MOVAS-1 cells. MOVAS-1 cells were incubated with YOH (10 or 20 μM) for 45 min and then treated with PDGF-BB (10 ng/mL) for 10 min. The protein expression levels of (**A**) phospho-ERK1/2, (**B**) phospho-p38, (**C**) phospho-AKT, (**D**) phospho-PLCγ1, and (**E**) phospho-(Ser) PKC substrates were determined through Western blotting. ** *p* < 0.01, *** *p* < 0.001 compared with the resting group; ^#^
*p* < 0.05 compared with the positive group. Data are presented as the mean ± S.E.M. (*n* = 3).

**Figure 5 ijms-23-08049-f005:**
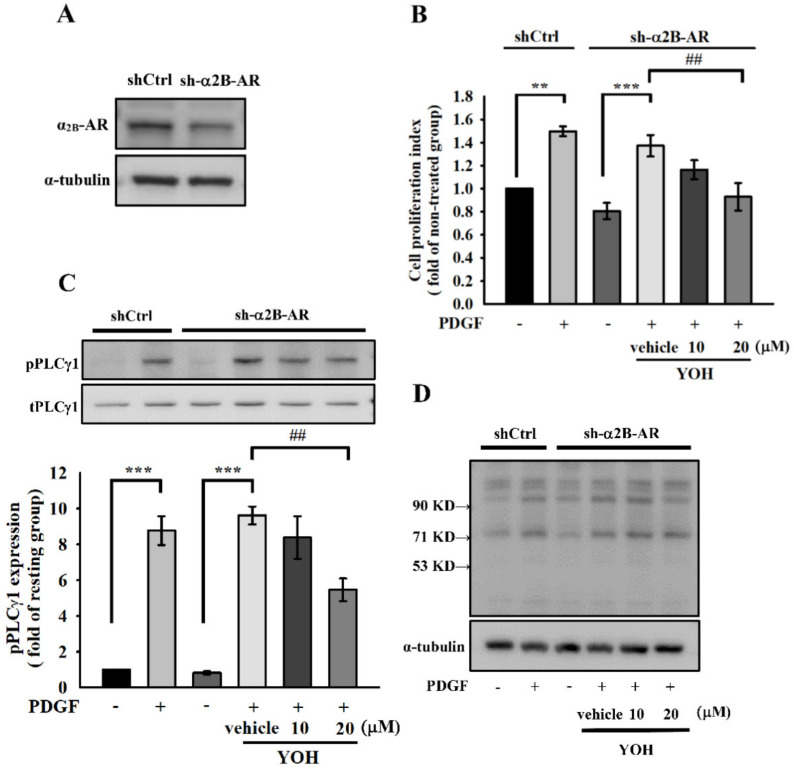
Effects of YOH on PDGF−BB-induced cell proliferation and PLCγ1 phosphorylation in MOVAS−1 cells with α2B−adrenergic receptor knockdown. We performed α2B−adrenergic receptor knockdown by incubating the cells with lentiviral particles of α2B−adrenergic receptor shRNA for 24 h and then culturing them with selection medium. (**A**) Expression of α2B−adrenergic receptors in wild−type cells and MOVAS−1 cells with α2B−adrenergic receptor knockdown. (**B**) Cell proliferation index was measured using the MTT assay, and protein expression levels of phospho−PLCγ1 (**C**) and phospho− (Ser) PKC substrates (**D**) were recognized by anti-phospho−PLCγ1 or anti−phospho− (Ser) PKC substrate antibodies, respectively, and then determined through Western blotting. ** *p* < 0.01, *** *p* < 0.001 compared with the non−PDGF-treated group; ^##^
*p* < 0.01 compared with the positive control group. Data are presented as ± S.E.M. (*n* = 3).

**Figure 6 ijms-23-08049-f006:**
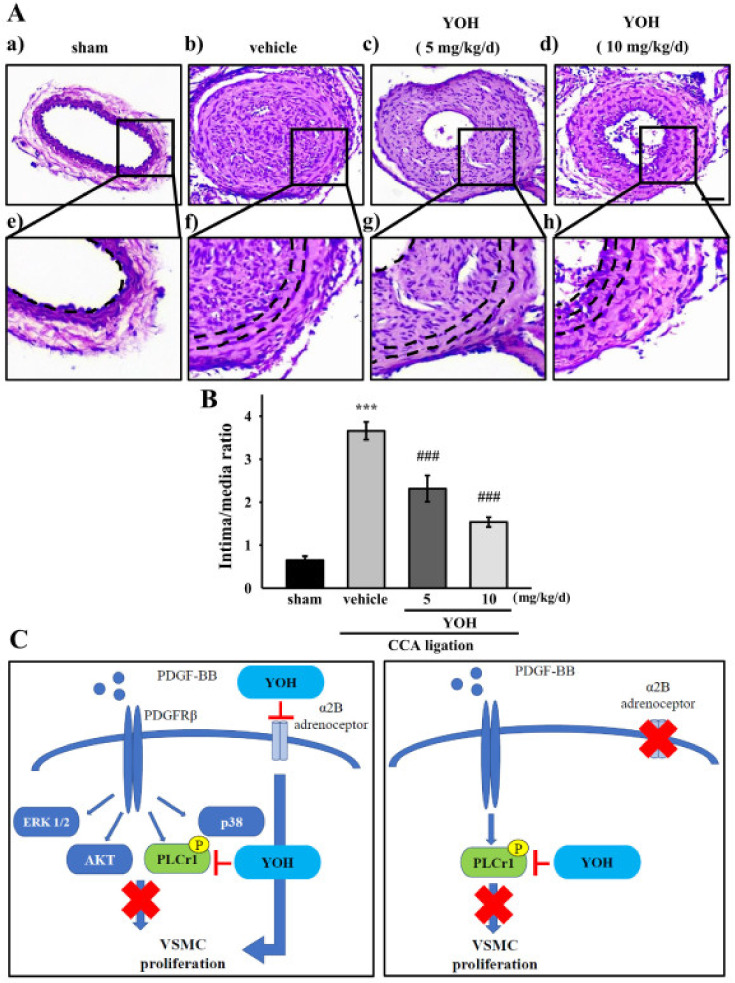
Effect of YOH on neointimal formation induced in mouse model of CCA ligation. (**A**) C57BL/6 mice were treated with vehicle or YOH (5 or 10 mg/kg) for 21 days after CCA ligation. The carotid arteries were harvested after 21 days. All CCA cross sections were stained with H&E; the representative patterns were obtained from five independent experiments. Arrowheads indicate elastic lamina, and the black bar indicates 100 mm. (**a**) sham, (**b**) vehicle, (**c**) YOH (5mg/kg/d), (**d**) YOH 10 mg/kg/d; (**e**–**h**) magnified parts of (**a**–**d**); (**B**) Quantifications of mean intimal thickness and intimal–medial layer ratio. Data are presented as the mean ± S.E.M. (*n* = 5 for each experimental group). *** *p* < 0.001 compared with the sham group; ^#^^##^
*p* < 0.001 compared with the injured group. (**C**) Schematic of the inhibitory mechanism of YOH in wild-type cells and VSMCs with α2B-adrenergic receptor knockdown. YOH inhibited PDGF-BB-induced proliferation by suppressing PLCγ1 phosphorylation but not ERK1/2, AKT, or p38. Similarly, YOH treatment still inhibited cell proliferation through PLCγ1 phosphorylation downregulation in PDGF-BB-stimulated VSMCs with α2B-adrenergic receptor knockdown.

## Data Availability

All data generated or analyzed during this study are included in this published article.

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
