# Peer review of "Yohimbine, an α2-Adrenoceptor Antagonist, Suppresses PDGF-BB-Stimulated Vascular Smooth Muscle Cell Proliferation by Downregulating the PLCγ1 Signaling Pathway"

_ijms, 2022, doi:10.3390/ijms23148049_

Round 1

Reviewer 1 Report

Chiu, Sheu and colleagues present here studies testing the effect of yohimbine hydrochloride on vascular smooth muscle cells that have been stimulated to proliferate with PDGF-BB, as a model for proliferation in disease. The majority of the study focusses on Western blot analyses of cell lysates, and investigating the effect of the yohimbine on protein levels or phosphoprotein levels, however other assays are used in support.  The basic story is interesting, and I believe that the experiments have been well conducted, however a significant number of presentation issues must be addressed before it can be published.

The authors refer to “VSMC” cells throughout the manuscript, which is ambiguous as it could refer to primary cells, or one of various cell lines from various species. They only clarify in the methods that they are using a specific mouse cell line, MOVAS-1 (at least as far as I can see).  The authors should be more explicit about this, and refer to the cell line rather than VSMC in all experimental contexts.

In the first paragraph of the results, the authors state that they perform “the MTT assay” but provide no detail or context to what this is.  Non-specialists would find this even more confusing as the assay in question is not referred to as the MTT assay in methods, but instead as the “Proliferation assay”.  Please provide a brief introduction to this assay in the results section to explain the context.

In figures such as 1Ba where different concentrations of yohimbine are displayed as numbers, I would like to see the unit added.

Fig 1A does not contain any labelling in figure to state what a, b, c and d display, only in the legend. Please add labels to the figure itself for clarity.

Similarly, in several of the Western blots (e.g. Fig 2Ba), there are no labels on the columns, again making it hard to interpret at a glance.

For the many blots looking at the effect of PDGF-BB and yohimbine (e.g. figures 3-5), were separate lysates prepared for each test, or were the same lysates tested on all of them? In the event that staining from the same membrane are shown, could figures be incorporated together

In a few of the blots (e.g. Fig 3Aa and 3Ab), the actin signal appears saturated.

Was actin used to normalize quantified Western blot data, to discount loading error?  For phospho-proteins (e.g. pAkt), was total protein (anti-Akt) also used to differentiate between increases in protein expression and increases in protein phosphorylation?

In figure 5, it appears that the shRNA is only used in the presence of PDGF-BB. Why Are PDGF-BB with the control shRNA and the  receptor shRNA without PDGF-BB not also included as controls?

Does the lysis buffer used contain enzyme inhibitors, and if so which ones?

Figure 5D does not state which antibody is used on it.

Minor issues

The abbreviation YOH is OK for yohimbine, but not for yohimbine hydrochloride, as it suggests a hydroxide rather than a hydrochloride.

I personally find the numbering of the figures with sub-sub-sections  (Fig 1Ac for example) as unnecessarily confusing, and would prefer it to be simplified. However, this is ultimately a choice for the authors and editors.

Author Response

Response to Reviewer #1

We appreciate your comments for this study and paid careful attention to it as we were revising our manuscript. The following is our responses to the specific issues raised.

Comments:

  1. The authors refer to “VSMC” cells throughout the manuscript, which is ambiguous as it could refer to primary cells, or one of various cell lines from various species. They only clarify in the methods that they are using a specific mouse cell line, MOVAS-1 (at least as far as I can see). The authors should be more explicit about this, and refer to the cell line rather than VSMC in all experimental contexts.

Response:

        We are thankful for your suggestion. “VSMC” has been changed to “MOVAS-1” in all experimental contexts. The changes in the newly revised manuscript were highlighted with red shading (lines 19, 20, 22, 26, 27, 85, 89, 90, 95, 97, 101, 108, 138, 142, 152, 157, 159, 160, 163, 164, 188, 199, 202, 205, 235, 244, 249, 251, 252, 254, 255, 284, 292, 297, 299, 301, 303, 330, 333, 334, 421, 430, 439, 448, 449, 472, 473, 525, 533, 540).

  1. In the first paragraph of the results, the authors state that they perform “the MTT assay” but provide no detail or context to what this is. Non-specialists would find this even more confusing as the assay in question is not referred to as the MTT assay in methods, but instead as the “Proliferation assay”. Please provide a brief introduction to this assay in the results section to explain the context.

Response:

A brief introduction of MTT assay has been added into the “Result” section of the newly revised manuscript (lines 87&88). Besides, the title of proliferation assay in the “Method” section was also modified (line 524).

  1. In figures such as 1Ba where different concentrations of yohimbine are displayed as numbers, I would like to see the unit added.

Response:

        The concentration unit was incorporated into new figures including Figs 1Ba&b, Fig 2Ae, Figs 3A&B, Figs 4A-E, and Figs 5B-D in the revised manuscript.

  1. Fig 1 does not contain any labelling in figure to state what a, b, c and d display, only in the legend. Please add labels to the figure itself for clarity.

Response:

        Thank you for the reminder. Different treatment of a-d groups was labeled in new Fig 1C of revised manuscript.

  1. Similarly, in several of the Western blots (e.g. Fig 2Ba), there are no labels on the columns, again making it hard to interpret at a glance.

Response:

        We are so sorry that me missed the labels of Fig 2Ba. The labeling of new Fig 2Ba was presented in the revised manuscript. In addition, since the labels were shown on the columns of statistical figures, the bands of western blots were not further labeled.    

  1. For the many blots looking at the effect of PDGF-BB and yohimbine (e.g. figures 3-5), were separate lysates prepared for each test, or were the same lysates tested on all of them? In the event that staining from the same membrane are shown, could figures be incorporated together

Response:

        All experiments in Figs 3-5 were performed three times, the protein cell lysates were purified and then underwent western blotting assay. Those data should be not from the same cell lysates. Usually, only single typical pattern will be presented. Regarding this issue, we have provided all 3 blots of each experiment in Figs 3-5 in the attached file. Hopefully, these raw data can clarify this issue.

  1. In a few of the blots (e.g. Fig 3Aa and 3Ab), the actin signal appears saturated.

Response:

        The actin signal of Figs 3Aa&b have been adjusted in the newly revised manuscript.

  1. Was actin used to normalize quantified Western blot data, to discount loading error? For phospho-proteins (e.g. pAkt), was total protein (anti-Akt) also used to differentiate between increases in protein expression and increases in protein phosphorylation?

Response:

        Total protein of Akt, p38, ERK, PLCg1 have been evaluated as internal control in the experiments of Fig 4. The new Fig 4 has been presented in the revised manuscript.  

  1. In figure 5, it appears that the shRNA is only used in the presence of PDGF-BB. Why Are PDGF-BB with the control shRNA and the receptor shRNA without PDGF-BB not also included as controls?

Response:

        Indeed, your design of this experiment is more comprehensive than ours. We did not conduct all control groups because we thought that it is easier for people to conclude the results directly. On the other hand, based on our experiences, we have known that the transfection of control and receptor shRNA did not affect cell viability as well as activity on MOVAS-1 cells. We thus did not provide more control groups in this experiment. Since we only have one week for the first revision, we thus cannot provide more control groups of this experiment this time. Although it does not affect the conclusion of this study, if you think it is necessary to add more control groups. We may need more time to perform the knockdown cells and complete this experiment.

  1. Does the lysis buffer used contain enzyme inhibitors, and if so which ones?

Response:

        The description of enzyme inhibitors was added in the “Method” section and highlighted with red shading (lines 548-550).

  1. Figure 5D does not state which antibody is used on it.

Response:

        The antibodies which used in Figure 5D were mentioned in the new legend of Figure 5D and highlighted with red shading (lines 336-337) in the revised manuscript.

  1. The abbreviation YOH is OK for yohimbine, but not for yohimbine hydrochloride, as it suggests a hydroxide rather than a hydrochloride.

Response:

We are sorry for this mistake. The abbreviation of yohimbine was corrected in the abstract and introduction sections (lines 15 & 63). Yohimbine hydrochloride was mentioned in the section of “Materials and Methods” (lines 492, 511-512).

We appreciate all your comments for this study, and hope the above changes meet with your agreement. Your suggestions truly improve this study. 

Reviewer 2 Report

 GENERAL COMMENTS 

The topic is interesting as it provides important insight into the signaling pathway involved in VSMC proliferation. In this sense, the manuscript addresses a worthwhile topic. However, some aspects might be improved.

The manuscript may benefit from considering the following aspects:

At the end of the Introduction: formulate the working hypothesis – what did you expect and why.

Discussion

Interestingly, while α(2)-autoinhibition is a key determinant of the magnitude of facilitation caused by angiotensin II in mesenteric vessels (ref Talaia C, Morato M, Quintas C, Gonçalves J, Queiroz G. Functional crosstalk of prejunctional receptors on the modulation of noradrenaline release in mesenteric vessels: A differential study of artery and vein. Eur J Pharmacol. 2011 Feb 10;652(1-3):33-9), leptin blocks the vasoconstrictor action of angiotensin II and inhibits the angiotensin II-induced increase in intracellular Ca(2+) in VSMCs (ref Fortuño A, Rodríguez A, Gómez-Ambrosi J, Muñiz P, et al. Leptin inhibits angiotensin II-induced intracellular calcium increase and vasoconstriction in the rat aorta. Endocrinology. 2002 Sep;143(9):3555-60). These additional intracelular mechanisms can be mentioned in the Discussion.

In the Discussion, it would be interesting to provide some more translational perspective. In this sense, yohimbine has been shown to decrease body fat (refs  Greenway FL, Bray GA, Heber D. Topical fat reduction. Obes Res. 1995 Nov;3 Suppl 4:561S-568S  //  Dudek M, Knutelska J, Bednarski M, NowiÅ„ski L, Zygmunt M, Mordyl B, GÅ‚uch-Lutwin M, Kazek G, Sapa J, Pytka K. A Comparison of the Anorectic Effect and Safety of the Alpha2-Adrenoceptor Ligands Guanfacine and Yohimbine in Rats with Diet-Induced Obesity. PLoS One. 2015 Oct 27;10(10):e0141327), which could be viewed as an added value given the relation between obesity and cancer. This aspect could be mentioned.

Author Response

Response to Reviewer #2

We appreciate your comments for this study and paid careful attention to it as we were revising our manuscript. The following is our responses to the specific issues raised.

Comments.

  1. At the end of the Introduction: formulate the working hypothesis – what did you expect and why.

Response:

We are thankful for your suggestion. The hypothesis and expectation were rewritten in the end of “Introduction” section. We have highlighted the changes in the newly revised manuscript with red shading (lines 75-82).

  1. Discussion

Interestingly, while α(2)-autoinhibition is a key determinant of the magnitude of facilitation caused by angiotensin II in mesenteric vessels (ref Talaia C, Morato M, Quintas C, Gonçalves J, Queiroz G. Functional crosstalk of prejunctional receptors on the modulation of noradrenaline release in mesenteric vessels: A differential study of artery and vein. Eur J Pharmacol. 2011 Feb 10;652(1-3):33-9), leptin blocks the vasoconstrictor action of angiotensin II and inhibits the angiotensin II-induced increase in intracellular Ca(2+) in VSMCs (ref Fortuño A, Rodríguez A, Gómez-Ambrosi J, Muñiz P, et al. Leptin inhibits angiotensin II-induced intracellular calcium increase and vasoconstriction in the rat aorta. Endocrinology. 2002 Sep;143(9):3555-60). These additional intracelular mechanisms can be mentioned in the Discussion.

Response:

        We truly appreciate your suggestion. These references and discussion of vasoconstriction were added into the “Discussion” section (lines 477-482) with red shading in the newly revised manuscript.

  1. In the Discussion, it would be interesting to provide some more translational perspective. In this sense, yohimbine has been shown to decrease body fat (refs Greenway FL, Bray GA, Heber D. Topical fat reduction. Obes Res. 1995 Nov;3 Suppl 4:561S-568S //  Dudek M, Knutelska J, Bednarski M, NowiÅ„ski L, Zygmunt M, Mordyl B, GÅ‚uch-Lutwin M, Kazek G, Sapa J, Pytka K. A Comparison of the Anorectic Effect and Safety of the Alpha2-Adrenoceptor Ligands Guanfacine and Yohimbine in Rats with Diet-Induced Obesity. PLoS One. 2015 Oct 27;10(10):e0141327), which could be viewed as an added value given the relation between obesity and cancer. This aspect could be mentioned.

Response:

These references and discussion of lipolysis and yohimbine were also added into the “Discussion” section (lines 404-407) with red shading in the newly revised manuscript.

We appreciate your comments for this study, and hope the above changes meet with your agreement. Your comments and suggestions truly improve the discussion of our study.   

Round 2

Reviewer 1 Report

The authors have addressed the vast majority of my concerns. The only one not addressed is listed below, which the authors say that they have not done because the one week revision period provided was not long enough to conduct this experiment.

In figure 5, it appears that the shRNA is only used in the presence of PDGF-BB. Why Are PDGF-BB with the control shRNA and the  receptor shRNA without PDGF-BB not also included as controls?

I therefore ask the journal to consult with the authors on how much time would be required, and to provide if for them.  Once these additional controls are conducted, and assuming that the results are as expected, then this paper will be suitable for publication.

Author Response

We appreciate your comment for this study again and paid careful attention to it as we were revising our manuscript. The following is our response to the specific issue raised.

Comment:

In figure 5, it appears that the shRNA is only used in the presence of PDGF-BB. Why Are PDGF-BB with the control shRNA and the receptor shRNA without PDGF-BB not also included as controls?

Response:

We have added new Fig. 5B, C, D with more control groups as you asked in the newly revised manuscript, and figure legend of Fig. 5 have also been modified and highlighted with red shading (line. 338, 339).

We appreciate your comment for this study, and hope the above changes meet with your agreement. Your truly improve this study.
